Title
**Eddy Covariance Evaluation of Ecosystem Fluxes at a Temperate Saltmarsh in**
**Victoria, Australia Shows Large CO$_2$ Uptake**
Authors
Ruth Reef[1],
Edoardo Daly[2,3],
Tivanka Anandappa[1],
Eboni-Jane Vienna-Hallam[1],
Harriet Robertson[1],
Matthew Peck[1],
Adrien Guyot[4,5]
Affiliations
1 School of Earth, Atmosphere and Environment, Monash University, VIC 3800, Australia
2 Department of Civil Engineering, Monash University, VIC 3800, Australia
3 WMAwater, Brisbane, QLD 4000, Australia
4 Atmospheric Observations Research Group, The University of Queensland, Brisbane,
Australia
5 Australian Bureau of Meteorology, Melbourne, Australia
Corresponding Author
Associate Professor Ruth Reef
School of Earth Atmosphere and Environment
Monash University
9 Rainforest Walk, Clayton VIC 3800
Australia
Email: ruth.reef@monash.edu
Ph: +61 3 9905 8309


Key Points

This is the first study using eddy covariance to measure $CO_2$ fluxes at an Australian
temperate saltmarsh, revealing temperature and light limitations to $CO_2$ uptake.

$CO_2$ fluxes varied seasonally; growing season net ecosystem productivity was 10.54 g $CO_2$
$m^{-2}$ $day^{-1}$, dropping to 1.64 g $CO_2$ $m^{-2}$ $day^{-1}$ in winter.

Productivity at the French Island saltmarsh is high relative to global saltmarsh estimates but
below global mangrove averages.



Abstract

Recent studies highlight the important role of vegetated coastal ecosystems in atmospheric
carbon sequestration. Saltmarshes constitute 30% of these ecosystems globally and are the
primary intertidal coastal wetland habitat outside the tropics. Eddy covariance (EC) is the
main method for measuring biosphere-atmosphere fluxes, but its use in coastal environments
is rare. At an Australian temperate saltmarsh site on French Island, Victoria, we measured
$CO_2$ and water gas concentration gradients, temperature, wind speed and radiation. The
marsh was dominated by a dense cover of *Sarcocornia quinqueflora*. Fluxes were seasonal,
with minima in winter when vegetation is dormant. Net ecosystem productivity (NEP) during
the growing season averaged 10.54 g $CO_2$ $m^{-2}$ $day^{-1}$ decreasing to 1.64 g $CO_2$ $m^{-2}$ $day^{-1}$ in
the dormant period, yet the marsh remained a $CO_2$ sink due to some sempervirent species.
Ecosystem respiration rates were lower during the dormant period compared with the
growing season (1.00 vs 1.77 μmol $CO_2$ $m^{-2}$ $s^{-1}$) with a slight positive relationship with
temperature. During the growing season, fluxes were significantly influenced by light levels,
ambient temperatures and humidity with cool temperatures and cloud cover limiting NEP.
Ecosystem water use efficiency of 0.86 g C $kg^{-1}$ $H_2O$ was similar to other C3 intertidal
marshes and evapotranspiration averaged 2.48 mm $day^{-1}$ during the growing season.

EGUsphere Topics
Emissions, Marine and Freshwater Biogeosciences, Earth System Biogeosciences

Short Summary

Studies show that saltmarshes excel at capturing carbon from the atmosphere. In this study,
we measured $CO_2$ flux in an Australian temperate saltmarsh on French Island. The temperate
saltmarsh exhibited strong seasonality. During the warmer growing season, the saltmarsh
absorbed on average 10.5 grams of $CO_2$ from the atmosphere per $m^2$ daily. Even in winter,
when plants were dormant, it continued to be a $CO_2$ sink, albeit smaller. Cool temperatures
and high cloud cover inhibit carbon sequestration.




## 1. Introduction

Despite their relatively small global footprint of 54,650 km² (Mcowen et al., 2017), salt marshes provide a range of ecosystem services, including shoreline protection (Shepard et al., 2011), nutrient uptake, nursery grounds for fish populations (Whitfield, 2017) as well as functioning as significant carbon sinks through $CO_2$ uptake and storage in their organic rich sediments (McLeod et al., 2011). These 'blue carbon' habitats are recognised for their significant contribution to the global carbon cycle, as coastal wetlands more broadly are estimated to have accumulated more than a quarter of global organic soil carbon (Duarte, 2017).

Saltmarshes are a widely distributed intertidal habitat but are floristically divergent globally (Adam, 2002), such that commonalities in function and form do not extend across biogeographic realms. US saltmarshes, for example, are extensively dominated by a single grassy species, *Spartina alterniflora*, as opposed to the dominance of $C_3$ Chenopodioideae species in the southern hemisphere (Adam, 2002). Temperate saltmarshes occupy a latitudinal range spanning from approximately 30° to 60° (Mcowen et al., 2017) and are most commonly found along protected coastlines such as bays, estuaries, and lagoons, where they are sheltered from the full force of wave action (Mitsch and Gosselink, 2000). In the Southern Hemisphere, temperate saltmarshes have a strong Gondwanan element with high floristic similarity among the marshes of New Zealand, the southernmost coasts of South America and South Africa and the southern coastlines of Australia (Adam, 1990). These marshes are often associated with extensive seagrass meadows and mudflats, and in parts of their range, mangroves, forming complex coastal mosaics (Huxham et al., 2018). Saltmarshes have been heavily degraded across their range, and it is estimated that perhaps up to 50% of the global saltmarsh area has been lost since 1900 (Gedan et al., 2009), primarily due to land use change.

In most areas where they occur, seasonality plays a major role in the functioning of temperate saltmarshes (Ghosh and Mishra, 2017). These ecosystems experience distinct growing and dormant seasons, primarily driven by temperature, light availability, and precipitation patterns (Adam, 2000). During the growing season (typically spring and summer), increased temperatures and longer daylight hours stimulate plant growth, photosynthetic activity, and

decomposition processes. Photosynthesis typically outpaces decomposition during this
period, resulting in the temperate saltmarsh acting as a net $CO_2$ sink (Chmura et al., 2003).
Conversely, the dormant season (usually fall and winter) is characterized by cooler
temperatures and shorter days (Adam, 2000; Howe et al., 2010). These factors lead to
reduced plant growth and photosynthetic activity (Adam, 2000) and while decomposition
processes also slow down due to cooler temperatures, $CO_2$ release through decomposition
often exceeds $CO_2$ uptake during this period (Artigas et al., 2015). In Australia, saltmarshes
have been assumed to not exhibit seasonality (Owers et al., 2018) despite there being a
scarcity of data on saltmarsh phenology and the implication this untested assumption could
have on carbon budget estimations.

Gross primary production (GPP) of saltmarshes is the total amount of $CO_2$ uptake by plants
through photosynthesis. Respiration ($R_e$) leads to a $CO_2$ flux directed back to the atmosphere
due to all respiration processes occurring within the saltmarsh, involving both autotrophs and
heterotrophs. The difference between these two fluxes is the net ecosystem exchange (NEE).
Saltmarsh ecosystems can act as both sources and sinks of carbon dioxide ($CO_2$), influencing
atmospheric $CO_2$ concentrations (Chmura et al., 2003). However, quantifying their net
exchange remains challenging (Lu et al., 2017) hindering their effective inclusion in Earth
System Models (Ward et al., 2020) and confounding the incorporation of saltmarsh
restoration in emission reduction targets. Eddy covariance (EC) provides a powerful method
for near-continuous, high-frequency monitoring of gas exchange between a vegetated surface
and the atmosphere (Baldocchi, 2003), enabling the determination of net ecosystem exchange
(NEE) of $CO_2$, and identifying the forcings that determine how $CO_2$ fluxes will respond to
global climate change (Borges et al., 2006; Cai, 2011).

Previous EC studies in coastal saltmarshes have been focused on the Northern Hemisphere, in
sites in the USA (e.g. Hill and Vargas, 2022; Kathilankal et al., 2008; Moffett et al., 2010;
Nahrawi et al., 2020; Schäfer et al., 2019), France (Mayen et al., 2024), Japan (Otani and
Endo, 2019) and China (Wei et al., 2020) but interest in the southern hemisphere is growing
(Bautista et al., 2023). The NEE values from these studies indicate that there is high inter-site
(as well as interannual, Erickson et al., (2013)) variability in carbon dynamics of saltmarshes,
with a link to species types, salinity, hydrology (Moffett et al., 2010; Nahrawi et al., 2020),
site specific biochemical conditions (Seyfferth et al., 2020) and latitude (Feagin et al., 2020).
While generally considered important carbon sinks (e.g. ranging between 130 to 775 g C m$^{-2}$
yr$^{-1}$ in the USA, according to Kathilankal et al. (2008) and Wang et al,(2016) respectively)
and globally hypothesised to average 382 g C m$^{-2}$ y$^{-1}$ (Alongi, 2020), some EC studies
revealed saltmarshes to be net sources of $CO_2$ to the atmosphere (Vázquez-Lule and Vargas,
2021) especially in temperate saltmarshes that experience long dormant periods.

The aim of this study is to estimate $CO_2$ and water fluxes in a temperate saltmarsh in
Victoria, southern Australia, to better characterise the effect of seasonality and environmental
variables on the saltmarsh $CO_2$ budgets. This is the first study in an Australian coastal
saltmarsh where $CO_2$ fluxes are estimated using the EC method.

2.   Methods

2.1 Site Description

Ecosystem flux measurements were collected at the Tortoise Head Ramsar coastal wetland on
French Island, Victoria (38.388°S, 145.278°E, Fig. 1) within the Western Port embayment.
French Island is within the Cfb climate zone (temperate oceanic climate) and experiences
distinct seasonal variations in temperature and precipitation. Long term (30 year) climate data
averaged from the nearby Cerberus Station (Australian Bureau of Meteorology, site 86361)
indicated that summers, spanning from December through February, are generally mild to
warm, with maximum temperatures typically ranging from 17°C to 25°C although occasional
heatwaves lead to temporary spikes in temperature that can exceed 30°C. Winters, from June
to September, are cooler, with maximum temperatures ranging between 7°C and 14°C and a
mean minimum temperature of 6°C. Frost is infrequent due to maritime influence, though
crisp mornings below 0°C occur 10% of the time in winter. Rainfall, evenly distributed
throughout the year, averages ca. 715 mm y$^{-1}$, although in 2020 the site had higher than
average rainfall (860 mm y$^{-1}$). The island is exposed to weather patterns influenced by the
Southern Ocean and Bass Strait, leading to occasional storm systems, particularly in winter,
bringing gusty winds and increased precipitation. Western Port has semi-diurnal tides with a
range of nearly 3 m, resulting in wide intertidal flats occupied by mangroves of the species
*Avicennia marina* and saltmarshes. The saltmarsh in this study experiences complex
hydrological conditions, and we found that inundation does not directly link to tides.

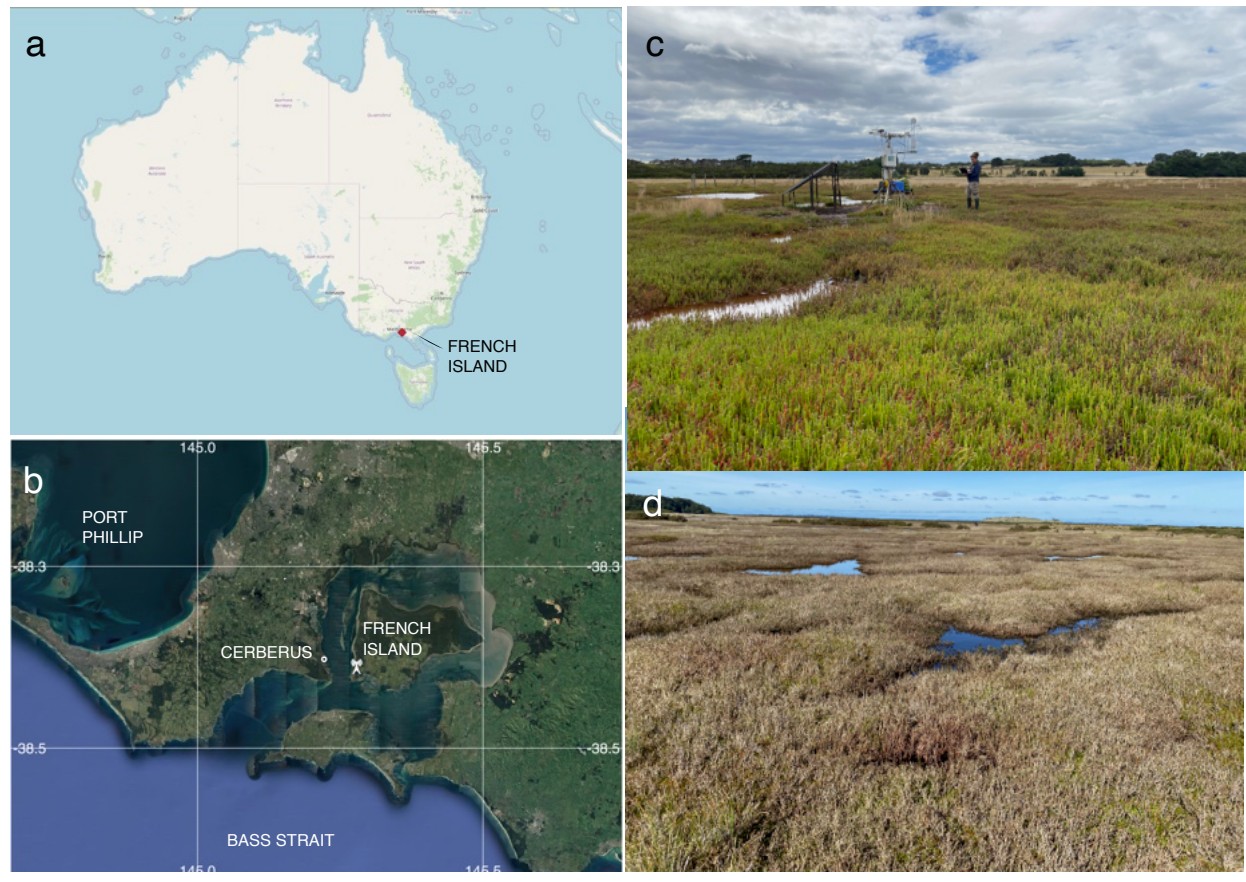

Figure 1: a) The location of French Island along the Bass Strait coast of Australia, and b) The location of the flux tower on French Island as well as the nearby Cerberus meteorological station (Bureau of Meteorology, Australia), © Google Earth. c) An image of the saltmarsh within the flux tower footprint during the growing season (with the tower and the author in the background), taken in February 2020 by Prudence Perry. d) an image of the saltmarsh during the dormant period, taken at the same location in September 2020 by Ruth Reef.

The site at French Island is dominated by an extensive temperate coastal saltmarsh community that is a particularly good natural representation of a broader biogeographic saltmarsh grouping which covers an area of ca. 7000 ha along Victoria's central coast embayments (Navarro et al., 2021). While the wetland at the site is a saltmarsh-mangrove-seagrass wetland system, the footprint of the flux tower was limited to the saltmarsh alone, which extends more than a kilometre from the shoreline in places. This geography provided the critical horizontally homogenous area with flat terrain required for ecosystem flux measurements. Floristically this saltmarsh is species poor, dominated by *Sarcocornia quinqeflora*. Stands of *Tecticornia arbuscula* are common in this saltmarsh, while *Atriplex cinerea*7pprox.7a*australia* and *Distichis distichophylla* can be prevalent depending on

elevation and soil drainage conditions. *Sarcocornia quinqeflora* is a perennial succulent and
at the temperate ranges of its distribution it has a distinct growing season from October to
May (Fig. 1c) when the stems turn red, followed by a woody and fibrous dormant period
during the colder months of June through September (Fig. 1d). The height of the dominant
vegetation ranged between 0.3 m.

2.2  Data Collection and Analysis

Eddy covariance measurements were made between November 2019 and August 2021
capturing both the saltmarsh growing season (October-May) as well as a dormant period
(June-September). An array of standard micro-meteorological instruments included a 3-
dimensional sonic anemometer (CSAT3, Campbell Scientific, USA), an open-path infra-red
carbon dioxide ($CO_2$) gas and water vapour ($H_2O$) analyser (Li-7500, Li-Cor, USA) and 2
data-loggers. The tower was powered by a solar array with two accompanying 12V DC
storage batteries. The sonic anemometer was mounted 2.3 m above ground. The CO2/H2O
gas analyser was mounted 0.11 m longitudinally displaced from the anemometer. A CR3000
datalogger (Campbell Scientific, USA), recorded the Li-7500, anemometer, short- and long-
wave radiation (CNR4, Klip & Zonen, the Netherlands), air temperature and humidity (083E,
Met One, USA) readings at 10 Hz frequency. Due to the location of the site in the Bass Strait
(a region that experiences regular winter storms, high wind speeds and higher than national
average cloud cover) the tower sustained damage due to winter storms several times during
the deployment, as well as suffered periods of poor power supply due to short day lengths
and high cloud cover; this was exacerbated by poor accessibility to the remote location during
COVID-19 travel restrictions. The analysis thus focused on extended periods of continuous
daily records and periods with large gaps in the dataset were removed.

Ecosystem fluxes were calculated for 30 min intervals using Eddy Pro software v.7 (LI-COR
Inc., USA) Express Mode protocols (see settings at
https://www.licor.com/env/support/EddyPro/topics/express-defaults.html). This processing
step includes coordinate axis rotation correction, trend correction, data synchronisation,
statistical tests for quality, density corrections and spectrum corrections. As part of this step,
flux quality flags were assigned to the calculated $CO_2$ fluxes using the 0–2 flag policy
'Mauder and Foken 2004', based on the steady state test and the developed turbulent

conditions test. The steady state test checks if fluxes remain consistent over the 30-minute averaging period by comparing the mean and standard deviation (SD) of fluxes in the first and second halves of the period. The developed turbulent conditions test ensures turbulence is well-developed and its energy spectra fits the Kolmogorov spectrum. Both tests assign partial flags that are combined into a single flag (0–2) in Eddy Pro, indicating the overall data quality. Only data that met the criteria of being in quality class 0 ('best quality fluxes') for $CO_2$ flux were chosen for further analysis. We further removed anomalous data points defined as values that exceed four SDs from the mean $CO_2$ flux; this resulted in the additional loss of ca. 1% of the dataset. Gap filling was not applied. Additional filtering was applied to nighttime data due to known weak convection at night, thus $CO_2$ flux data during periods of atmospheric stability, i.e. when night friction wind velocities (u*) were below 0.2 m s$^{-1}$, were excluded following inspection of the nightly NEE vs. u* curve to detect the threshold where NEE fall-off occurs (i.e. the Change Point Detection method, Barr et al., 2013). This resulted in a dataset of 674 day-time and 606 nighttime flux measurements during the dormant period and 4124 day-time and 3020 nighttime flux measurements for the growing period (Table 1). The growing season dataset included 90 days with 85% or more flux data coverage, while the dormant season dataset included 18 days, and these days were used for 24-hour flux integrations.

Table 1: Mean (±SD) net ecosystem exchange (μmol $CO_2$ m$^{-2}$ s$^{-1}$) during day- and nighttime respectively, as well as the corresponding number of half hourly measurements from each month, following filter applications (n).

| Month | Daytime Mean NEE (SD); n | Nighttime Mean NEE (SD); n | Season |
|---|---|---|---|
| October 2019 | -2.29 (3.08); 121 | 2.04 (1.28); 70 | Greening up |
| November 2019 | -1.84 (3.89); 151 | 2.85 (1.75); 110 | Greening up |
| December 2019 | -3.33 (4.59); 96 | 1.14 (1.70); 15 | Growing |
| January 2020 | -1.31 (3.31); 63 | 2.10 (0.79); 27 | Growing |
| February 2020 | -3.83 (4.11); 540 | 1.89 (1.10); 280 | Growing |
| March 2020 | -3.86 (3.90); 494 | 1.63 (0.78); 351 | Growing |
| August 2020 | 0.05 (2.05); 150 | 1.76 (1.22); 39 | Dormant |
| September 2020 | -0.98 (2.04); 147 | 1.27 (0.96); 101 | Dormant |
| January 2021 | -4.81 (5.04); 602 | 2.15 (1.55); 373 | Growing |
| February 2021 | -3.62 (4.27); 615 | 2.00 (1.19); 423 | Growing |
| March 2021 | -3.07 (3.95); 660 | 1.76 (1.20); 556 | Growing |

| | | | |
|---|---|---|---|
| April 2021 | -2.08 (3.02); 409 | 1.15 (0.87); 403 | Growing |
| May 2021 | -0.98 (2.57); 377 | 1.14 (1.04); 423 | End of Growing |
| June 2021 | 0.58 (1.67); 271 | 0.93 (1.30); 328 | Dormant |
| July 2021 | 1.07 (1.38); 102 | 0.82 (0.62); 127 | Dormant |



Half-hourly average $CO_2$ flux was measured in µmol m$^{-2}$ s$^{-1}$, with positive fluxes indicating a
flux direction from the Earth's surface to the atmosphere. Net ecosystem exchange (NEE)
was defined as the net flux of $CO_2$ from the atmosphere to the marsh and was often negative
during daytime, indicating that Gross Primary Productivity (GPP) was larger than ecosystem
respiration ($R_e$). Evapotranspiration (ET) was calculated by Eddy Pro as the ratio between the
latent heat flux (LE) and latent heat of vaporisation ($\lambda$). Ecosystem water use efficiency
(WUEe) was then expressed as the ratio between daytime net ecosystem productivity in g
$CO_2$ m$^{-2}$ h$^{-1}$ and evapotranspiration in mm h$^{-1}$.

A two-dimensional footprint estimation was provided according to the simple footprint
parameterisation described in Kljun et al. (2015) calculating the ground position of the
cumulative fraction of flux source contribution by distance for each 30-minute interval. We
assessed the short-term effects of environmental factors on $CO_2$ fluxes at a half-hourly time
scale (e.g. the effects of light, air temperature and vapour pressure deficit) using a series of
non-linear or linear models. These analyses were limited to the growing season, when the
plants were actively photosynthesising. To calculate the daily-integrated $CO_2$ and $H_2O$ fluxes,
the daily sum of these fluxes was determined for days with at least 85% data coverage. This
involved using the trapezoid rule to estimate the area under the curve for each of these 24-
hour periods. The trapezoid rule approximates the total flux by dividing the day into smaller
intervals, each lasting 1,800 seconds (30 minutes). For each data interval, the area is
calculated by averaging the flux values at the beginning and end of the interval, then
multiplying by the interval duration. These areas are then summed to obtain the total daily
flux. This method ensures that even with some missing data points, a reliable estimate of the
daily flux can be obtained. All post-processing and statistical analyses were performed in R
4.3.2 (R Core Team, 2024) including the packages *ggplto2*, *clifro*, *MASS*, *dismo*, *amerifluxr*,
*rmarkdown*, *geosphere*, *ggmap* and *gbm*.

For the $CO_2$ budget, Net Ecosystem Production (NEP), was defined as NEP=-NEE.
Nighttime NEE is referred to as $R_e$ and was corrected for temperature effects on respiration
using an exponential Arrhenius-type relationship (Lloyd and Taylor, 1994).

3   Results

The observations were divided into a growing season and a dormant season to reflect the
seasonal phenology of the dominant vegetation type within the flux tower footprint. During
the growing season, mean temperature averaged 22.3°C. Several heatwaves occurred during
this period, with temperatures exceeding 40°C on a few occasions in 2019. The dormant
season was significantly colder and windier, with frequent southerly winds (Fig. 2a).
Footprint models showed a slight variation in flux source between the two seasons, although
in both cases the size of the footprint and the vegetation composition within the footprint was
similar (Figs. 2b and 2c), but the shape was skewed to the north during winter due to the
prevalent southerly winds in that season (Fig. 2a). 70% of the flux measurement source was
from within 50 m of the tower, while the maximum length of the source location was 73 m.

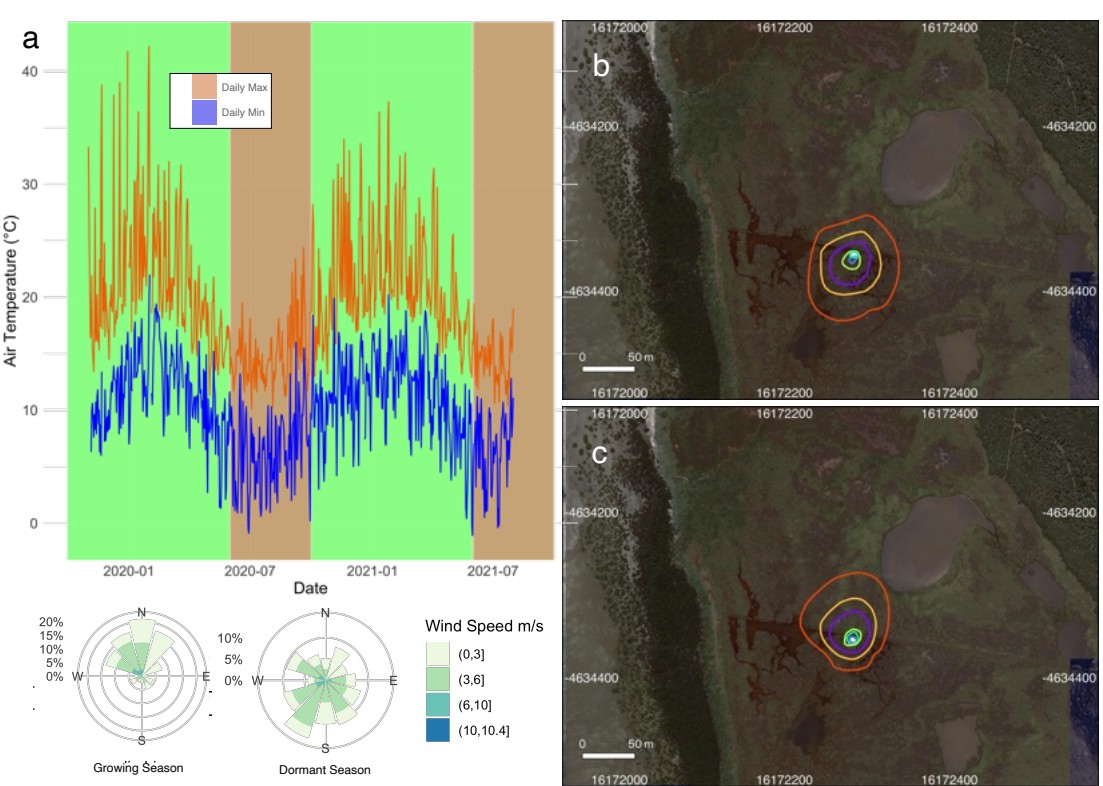




Figure 2: a) The minimum and maximum daily temperature recorded at the Cerberus meteorological station (Bureau of Meteorology, Fig. 1b) during 2019-2021. The marsh growing (October-May) and dormant (June-September) periods are shaded in green and pink respectively. A corresponding wind rose diagram summarises the wind speeds and directions measured at the tower site during the observation periods. The flux source footprint surrounding the tower during the dormant season (b) and the growing season (c) shows the cumulative flux source contribution to the flux measurements, with the outer red line representing the distance by which 90% of the calculated flux is sourced and the other isolines from the tower outwards correspond to 10%, 20%, 40%, 60% and 80% of the flux.

The growing season dataset included 90 days with 85% or more flux data coverage, while the dormant season dataset included 18 days. There was a strong temporal variability in net ecosystem exchange (NEE) across both short (daily) and long (seasonal) temporal scales (Fig. 3). Daytime fluxes were defined as flux points where the global radiation values in the flux averaging half-hour interval were >12 W m$^{-2}$ (as per EddyPro methodology). At the diurnal scale, saltmarsh NEE were negative mostly during the day and positive mostly during the night and ranged between -19.1 and 10.86 µmol m$^{-2}$ s$^{-1}$ across the measurement periods. Monthly averages and data coverage are shown in Table 1.


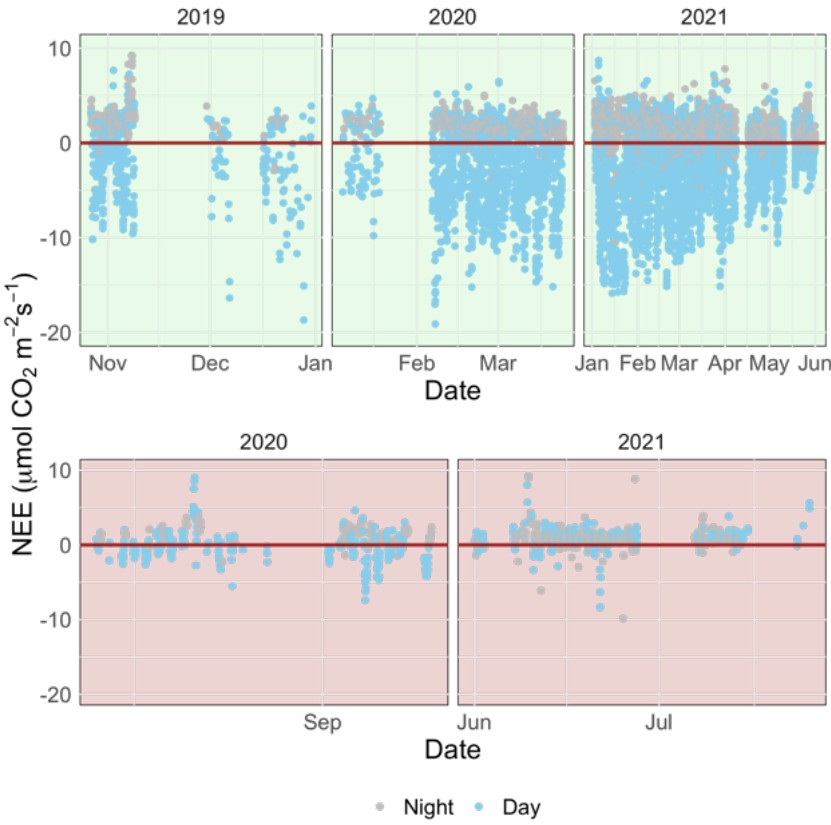


Figure 3: A time series of half-hourly measurements of $CO_2$ flux between a temperate
saltmarsh and the atmosphere measured by eddy covariance during the marsh growing season
(top) and the dormant season (bottom). Blue and grey points indicate measurements taken
during daytime and nighttime respectively. Positive fluxes indicate a direction of flux from
the Earth surface to the atmosphere.

Flux rates varied across the day, with $CO_2$ uptake peaking at 11:00 during the growing
season, and later in the day (14:00) during the dormant period (Fig. 4). Ecosystem respiration
rates ($R_e$, defined as nighttime $CO_2$ flux) were on average (±SD) 1.77 (±1.12) $\mu$mol m$^{-2}$ s$^{-1}$
during the growing season and 1.0 (± 0.93) $\mu$mol m$^{-2}$ s$^{-1}$ during the dormant period. The
difference in ecosystem respiration between the growing and dormant seasons is highly
significant (t-test, p<0.01). Daytime $CO_2$ flux was on average (±SD) -3.53 (± 4.15) $\mu$mol m$^{-2}$
s$^{-1}$ during the growing season and -0.25 (± 2.18) $\mu$mol m$^{-2}$ s$^{-1}$ during the dormant season.
Thus, we derive that the maximum Gross Primary Productivity (GPP) of this ecosystem from
NEE and temperature-corrected Re (Fig. 5), measured during the growing season, is ca. -5.34
± 4.3 $\mu$mol $CO_2$ m$^{-2}$ s$^{-1}$ (-5.53 ± 4.45 g C m$^{-2}$ day$^{-1}$). Average $R_e$ is thus estimated to comprise
33% of GPP.

Mean (±SD) daily evapotranspiration was 2.48 mm (±2.79 mm) during the growing season
and 0.97 mm (±1.35 mm) during the dormant season (Fig. 4). Evapotranspiration peaked at
noon AEST during the growing season (0.26 mm h$^{-1}$), and later in the day (14:00 AEST)
during the dormant season (0.14 mm h$^{-1}$).


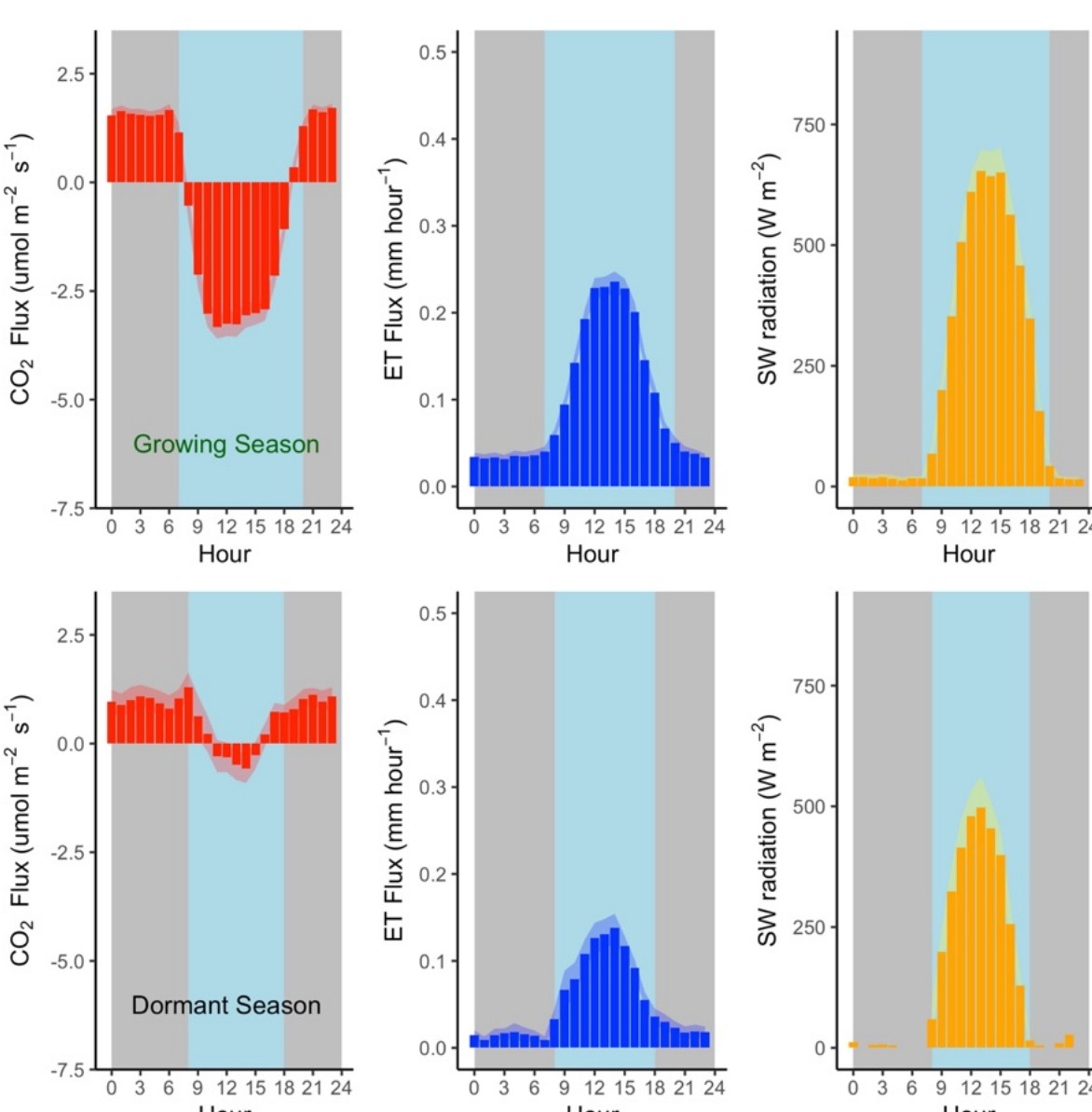



Figure 4: Mean hourly $CO_2$ and $H_2O$ flux (evapotranspiration) rates during the growing
season (top) and the dormant season (bottom) alongside mean short wave incoming radiation.
Shading corresponds to 1 standard deviation (SD) around the mean. Grey plot background

approximates nighttime periods, while light blue approximates daytime (actual day length
varies within each season).

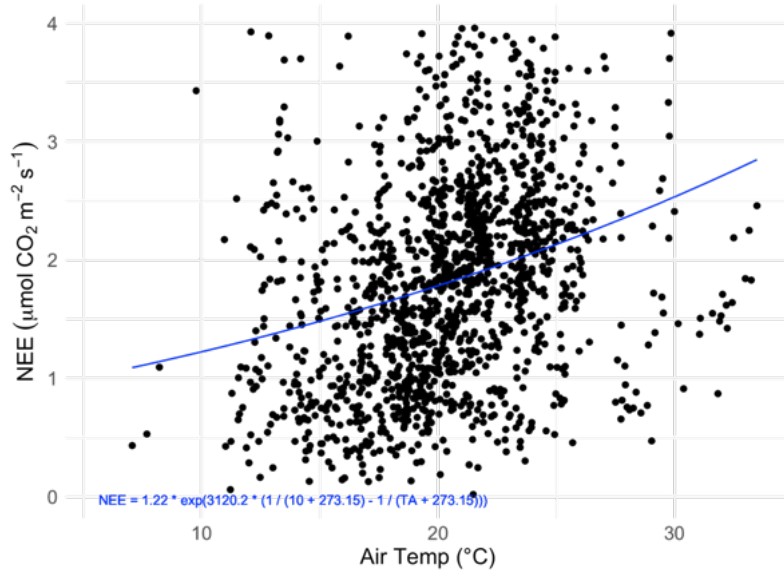

Figure 5: The relationship between nighttime half-hourly flux measurements (NEE) taken
between the hours of 22:00 and 02:00 and air temperature (TA). The fitted curve (blue line) is
the fitted Lloyd & Taylor Arrhenius non-linear model: NEE = 1.22*exp(3120.2*(1/283.2-
1/(TA+273.2))), $R^2$ = 0.09.
The effect of some environmental forcings on daytime NEE during the saltmarsh growing
season were explored (Fig. 6). To distinguish this daytime-only value from the 24-hour
carbon balance integration, and to better highlight $CO_2$ uptake, NEP values are shown.
Short wave radiation (visible light) was a limiting factor to NEP below approximately 300 W
m$^{-2}$, but radiation did not reach damaging levels that would lead to a drop in NEP throughout
the measurement range, which reached a maximum level of ca. 800 W m$^{-2}$. Unlike light, the
NEP-air temperature relationship followed a Gaussian response, with the highest NEP
achieved at the optimal temperature of 25.3°C with a SD of 3.8°C followed by a decline in
$CO_2$ uptake by the marsh at higher temperatures. The minimum and maximum air
temperatures for which modelled NEP nears zero (defined here as 3 SDs from the mean) are
13.9°C and 36.7°C respectively. Temperature also had a slight but significant positive linear
relationship with ecosystem respiration (slope=0.07 µmol $CO_2$ m$^{-2}$ s$^{-1}$ °C$^{-1}$, p<0.01, data not
shown).

NEP was positively correlated with evapotranspiration during the growing season (Pearson r
= 0.59, Fig.6 C). The slope of the NEP/ET relationship was 20.0, indicating an ecosystem
water use efficiency (WUE$_e$) of 0.86 g C kg$^{-1}$ $H_2O$ (R$^2$ = 0.34, p<0.001). The response of
NEP to atmospheric vapour pressure deficit (VPD) fit a Gaussian relationship (the commonly
observed inverse U-shaped curve relationship in response to VPD in plants), with NEP
declining rapidly when VPD exceeded 2.39 kPa. The optimal range of VPD within which
NEP was maximised in this ecosystem was 1.92 kPa (±0.73 kPa).

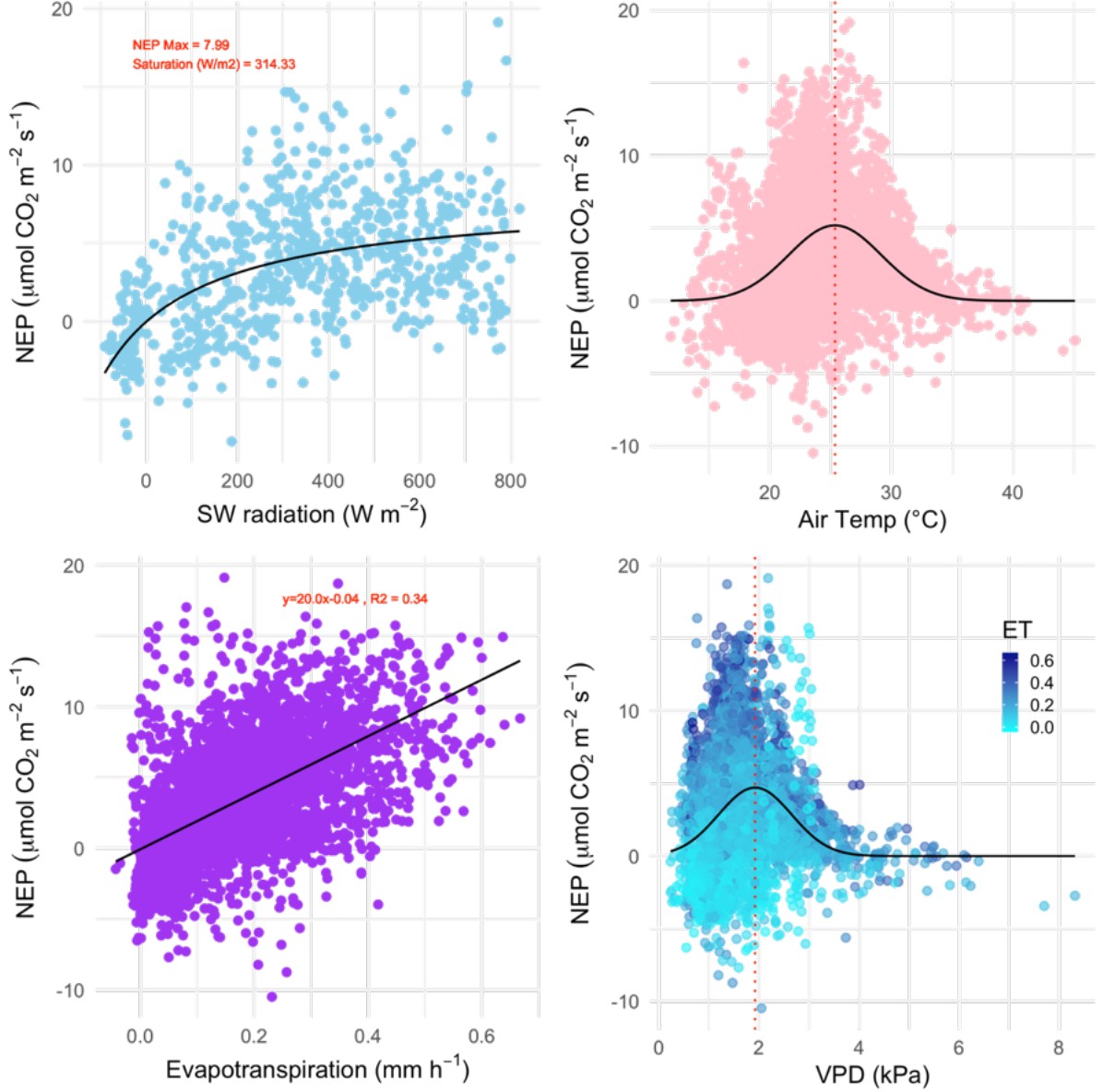


Figure 6: The relationship between growing season daytime half-hourly values of net
ecosystem productivity (NEP, $\mu$mol $CO_2$ $m^{-2}$ $s^{-1}$) and corresponding environmental variables.
a) Net shortwave (SW) radiation (visible light); black line is the Michaelis-Menten model of
best fit. The coefficient of saturation is at 314 W $m^{-2}$ and maximum net productivity is 8.0
$\mu$mol $CO_2$ $m^{-2}$ $s^{-1}$.  b) Air temperature (TA); black line is a Gaussian model of best fit with a
temperature optimum at 25.3°C. c) Evapotranspiration; linear model ($R^2 = 0.34$) has a slope
of 20.0. d) Vapour Pressure Deficit; black line is a Gaussian model of best fit with a VPD
optimum at 1.92 kPa, points are coloured by the level of evapotranspiration during the half
hourly NEP measurement.

When integrated over a 24-hour period, the saltmarsh is on average a daily $CO_2$ sink during
all canopy phenological phases (Fig. 7), although during the dormant season the sink is
weaker, with an average uptake of -2.42 g $CO_2$ $m^{-2}$ $day^{-1}$ (±2.54). During the growing season
(defined as the non-dormant period and thus reflecting several phenological stages), the
marsh is a substantial sink with a mean (±SD) daily NEP of 10.95 g $CO_2$ $m^{-2}$ $day^{-1}$ (±4.98)
over a 24-hour period (ranging between -22.8 and 4.3 g of $CO_2$ emission to the atmosphere
$m^{-2}$ $day^{-1}$) . The daily $CO_2$ budget during the growing season showed some variability among
days (CV=0.46, Fig. 7) and days with lower average light levels (i.e. cloudy days) had a
significant negative impact on the $CO_2$ budget (multiple linear regression, $p < 0.02$, $R^2 =$
0.27). Daily maximum air temperatures did not have a significant impact on the daily $CO_2$
budget ($p = 0.77$) at this location, although NEE was significantly affected by temperature at
finer temporal scales (Figure 6).


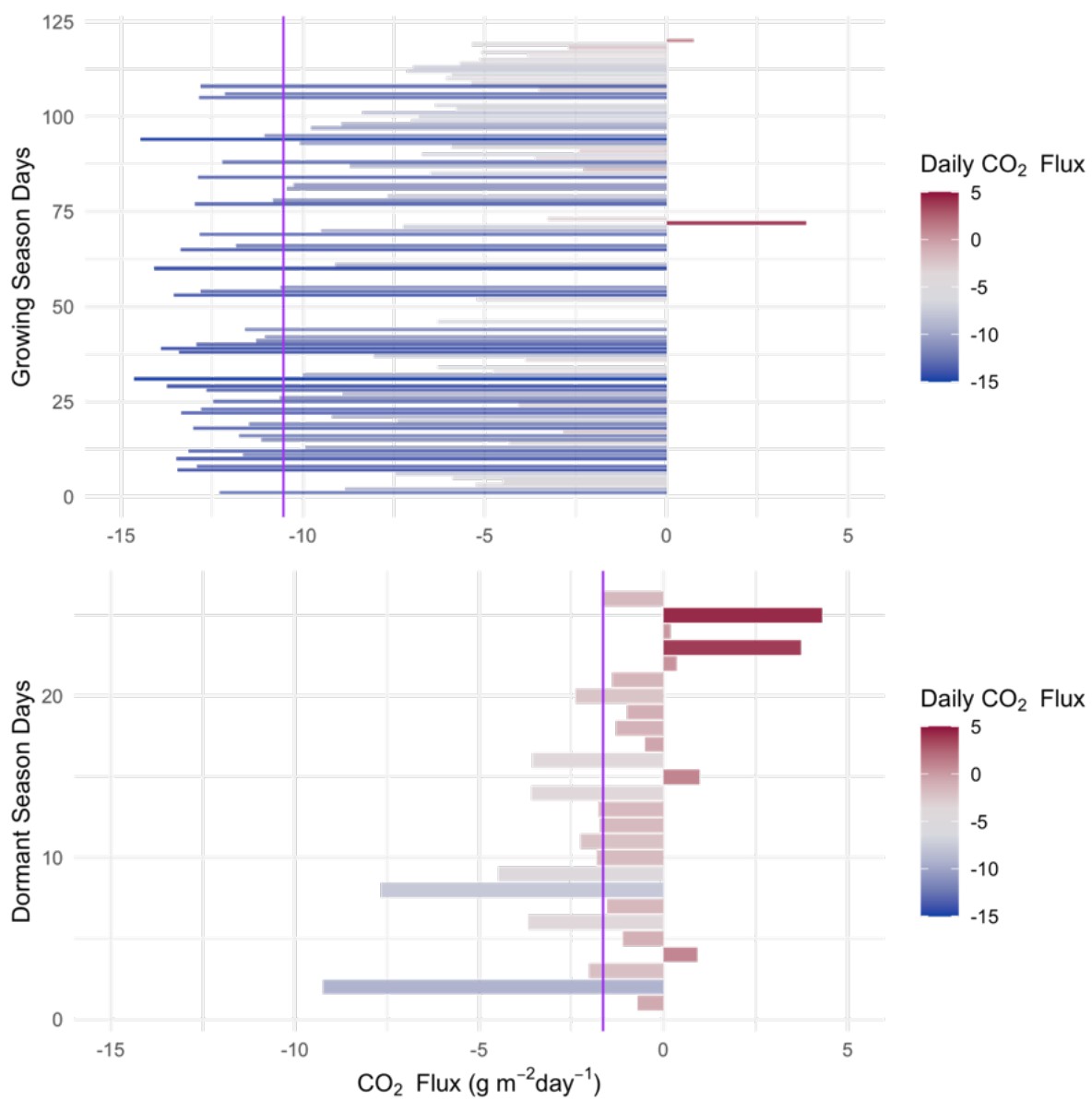


Figure 7: Daily (24 h) integrated NEE in g $CO_2$ m$^{-2}$ day$^{-1}$ during the saltmarsh growing
season (top) and the dormant season (bottom) for days with data density > 85%. Purple lines
indicate the mean daily integrated flux for each season (-10.54 and -1.64 g $CO_2$ m$^{-2}$ day$^{-1}$ with
an SD of 4.98 and 2.54 for growing and dormant respectively). A positive balance indicates
an integrated net flux of $CO_2$ from the Earth's surface to the atmosphere over the 24-hour
period.


4    Discussion

The study provided high-frequency measurements of an abundant greenhouse gas ($CO_2$)
using a precise technique (eddy covariance flux) in an ecosystem with limited historical
measurements. Time series analysis was performed on $CO_2$ flux measurements across various
scales (daily, nightly, diel, half-hourly, hourly, seasonally) to assess the impacts of ET, SW
radiation, VPD, and TA on $CO_2$ flux and how these relationships change throughout the year.
Seasonality was observed for the first time in an Australian saltmarsh and had a significant
effect on carbon and water flux. Growing season net ecosystem productivity was five times
greater than during the dormant period. Seasonality in Australian marshes has not been
previously reported in the scientific literature and contradicts previous assumptions that
Australian saltmarshes do not exhibit the growing and dormant phenology observed on other
continents (Clarke and Jacoby, 1994). Seasonality had a significant impact on the daily
carbon fluxes in this marsh and is an important characteristic of this habitat that has been
overlooked (Owers et al., 2018). Seasonality can also have other broader implications yet to
be considered in Australian marshes. For example, in the USA, the saltmarsh greening up
period was shown to be an important range-wide timing event for migratory birds (Smith et
al., 2020) with plant-growth metrics predicting the timing of nest initiation for shorebirds.
Saltmarshes in Australia are important roosting and feeding sites along the East Asian
Australasian Flyway, particularly for waders, thus potentially a similar relationship between
migration timing and saltmarsh phenology could be occurring. Seasonality also affects other
significant ecosystem functions such as the bio-geomorphological feedback between
saltmarshes, coastal hydrodynamics and landscape evolution (Reents et al., 2022).

We derived the light-response and associated coefficients of light regulation of saltmarsh
NEE using the Michaelis Menten model (Chen et al., 2002). Quantum (or production)
efficiency is the predominant input in remote sensing techniques to model productivity, and is
specific to the biome (Hilker et al., 2010). While not directly comparable to leaf level
quantum efficiency measurements, the quantum efficiency ($\alpha$) of the NEP light response
curve was estimated from the slope of the Michaelis-Menten model to be 0.025 $\mu$mol $CO_2$ $J^{-1}$.
The ecosystem reached light saturation at an insolation of 314 W $m^{-2}$, but daytime insolation
was below this value more than 50% of the time suggesting that light might be a significant
limiting factor to NEP at this marsh, especially during winter. The level of light limitation we
observed is an underestimation, due to the loss of high-quality EC data during periods of rain.
The solar geometry at this latitude and the length of day result in an annual average top of
atmosphere SW radiation of 250 W m$^{-2}$, but clouds can strongly modulate the SW radiation
balance (SWCRE), and apart from the months of January and February when cloudy days are
less frequent (10-12 days per month), cloudy days are frequent at this site, averaging 15-17
days per month (Bureau of Meteorology) and could significantly impact on NEP.

Temperature is another forcing that significantly impacts NEE at this marsh, with an optimal
range for maximum NEP at 25.3°C (21.5°C-29.1°C). Data for Australian saltmarshes is not
available, but this optimal temperature response range is similar to that measured
experimentally in a saltmarsh species in an equivalent climate zone (e.g. Georgia,
(Giurgevich and Dunn, 1981)) and to the values hypothesised for the habitat from data
collected along the US Atlantic Coast, (Feher et al., 2017). The long-term average maximum
daytime temperature at this site is 19.2°C, which is cooler than the optimal range for NEE
suggesting temperature can be a significant limiting factor to productivity, especially during
the dormancy period where average monthly maximum temperatures are only 13.7°C to
16.6°C (Bureau of Meteorology). During the growing season the average maximum
temperatures are within the range of optimal NEE (20.6°C to 23.1°C), although hot days
(>30°C) significantly depress NEE and depending on the year, can be common during
summer months (averaging 2-6 days per month). Within the diversity of saltmarsh species
found globally, some species have C4 photosynthetic pathways (Drake, 1989). C4
photosynthesis plants often exhibit higher optimum temperature ranges (30-35°C, Berry and
Björkman, 1980) than C3 photosynthesis plants (20-25), and the cooler conditions at this site
could explain the absence of C4 plants from this bioregion. The parabolic relationship
between NEP and air temperature and NEP and VPD suggest that higher air temperatures and
VPD (which are expected with climate change) could negatively impact $CO_2$ uptake by these
coastal ecosystems. High VPD was related to lower NEP, and to a lesser extent, lower ET
(Fig. 6d). However, VPD increases atmospheric demand for water, increasing the evaporation
from the saturated marsh surfaces in the footprint, and this atmospheric demand could be
forcing ET at high VPD rather than plant moderation via reduced transpiration, even if
transpiration is reduced. Thus, despite maintained ET during VPD periods we cannot
conclude a non-closure of stomata. NEP also reduced below a VPD of 1.92 KPa, but at our
field site low VPD correlated with low temperatures (r = 0.88), and low temperatures were
shown to limit NEP.

In saltmarshes, evapotranspiration occurs from plant mediated transpiration but also from soil
pores (which tend to be saturated), wetted leaves and open water. We observed average
evaporation rates of 2.48 mm day$^{-1}$ during the growing season and 0.97 mm day$^{-1}$ during the
dormant season. Actual evapotranspiration in this region modelled using the CMRSET
algorithm is estimated to range between 0.6 and 3.2 mm day$^{-1}$ during winter and summer
respectively (McVicar et al., 2022); our field measurements support the model. Overall,
rainfall is in excess of the requirements for maintaining ET at this site, although deficits can
develop for short periods during the growing season, when ET is higher, perhaps explaining
the drier saltmarsh surface during this period. Conversely, long term rainfall excess could be
contributing to the complicated hydrology at this location, where inundation is not strictly
associated with tidal stage (data not shown) and our observation of long (5-day) periods of
inundation during winter.

Growing season ET rates are significantly higher than those of the dormant season, partly due
to the solar configuration in winter as opposed to summer, but also due to phenological
changes. A big leaf model estimation of evapotranspiration from saltmarshes in New South
Wales estimates ET to be highly sensitive to vegetation height, increasing by more than 1 mm
day$^{-1}$ as vegetation height increases from 0.1 to 0.4 m (Hughes et al., 2001) and transpiration
in saltmarsh plants in the cold season has been shown to account for only 20% of the annual
transpiration budget (Giurgevich and Dunn, 1981) following the same pattern as the seasonal
distribution of productivity.

The rate of carbon uptake per unit of water loss (WUE) is a key ecosystem characteristic,
which is a result of a suite of physical and canopy physiological forcings, and has direct
implications for ecosystem function and global water and carbon cycling. Mean water use
efficiency (WUEe) of this saltmarsh was estimated at 0.86 g C kg$^{-1}$ H$_2$O, which is markedly
lower than for grass dominated saltmarshes in China (2.9 g C kg$^{-1}$ H$_2$O, Xiao et al. (2013))
but similar to the value for WUEe based on NEP and ET in mangroves (0.77 g C kg$^{-1}$ H$_2$O,
Krauss et al. (2022)), which are also C3 plants. The Chinese saltmarshes studied in Xiao et al.
(2013) are dominated by *Spartina alterniflora*, a C4 perennial grass. C4 plants have higher
(often double) water use efficiencies than C3 plants due to CO$_2$ concentrating mechanisms
(Osborne and Freckleton, 2009). The saltmarsh at French Island includes only C3 plants, and
the dominant chenopod *Sarcocornia quinqueflora* has been suspected to have higher
evapotranspiration rates than saltmarsh by approx. 15% (Hughes et al., 2001), but while
*Sarcocornia quinqueflora* dominates at this site, the footprint is a mix of species, and the
lower WUEe cannot be directly linked to the presence of *Sarcocornia quinqueflora*.
Furthermore, like most wetlands, the wetland surface is a mixed composition of emergent
vegetation, unsaturated soil and water bodies thus the spatial scale at which WUEe is
determined encompasses both the canopy (Ec) as well as any open water present in the
footprint. Transpiration is predicted to account for only 55% of ET in these systems (Hughes
et al., 2001), which is an Ec to ET ratio similar to that of mangroves (Krauss et al., 2022) but
significantly lower than terrestrial forests where more than 90% of ET can be attributed to
transpiration. Thus, regional variations in WUEe can be attributed to multiple forcings that
form complex spatiotemporal patterns.

Saltmarshes are considered among the most productive ecosystems on Earth with an
estimated global NEP of 634 Tg C $y^{-1}$ (Fagherazzi et al., 2013) and 601 634 Tg C $y^{-1}$
(Rosentreter et al., 2023). Productivity of southern Australian marshes was previously
estimated at 0.8 kg $m^{-2}$ $y^{-1}$ by repeated measurements of above ground standing crops (Clarke
and Jacoby, 1994), which if not accounting for season, equates to 2.2 g C $m^{-2}$ $d^{-1}$. Similar
studies on saltmarshes in France report lower productivity (483 g C $m^{-2}$ $y^{-1}$, (Mayen et al.,
2024)) and daily growing season rates of 1.53 g C $m^{-2}$ $d^{-1}$, but mid-latitude saltmarsh sites in
the USA and China show productivity rates of 775 g C $m^{-2}$ $y^{-1}$, (Wang et al., 2016) and 668 g
C $m^{-2}$ $y^{-1}$, (Xiao et al., 2013) respectively. It is clear that productivity across climate zones
and biogeographic regions varies widely with some studies even reporting net emissions over
an annual period from some marshes and a global average estimated between 382 (Alongi,
2020) and 1,585 g C $m^{-2}$ $y^{-1}$ (Chmura et al., 2003), albeit based on a small subset of studies.
An analysis of GPP across latitudes in the USA show that warmer sites (including mangrove
wetlands in southern USA) had significantly higher GPP than mid-latitude saltmarshes such
as the one on French Island (Feagin et al., 2020). Mangroves have higher NEE than
saltmarshes, estimated by Krauss et al. (2022) to average 1200 g C $m^{-2}$ $y^{-1}$. While our data
does not provide enough coverage for a long-term annual estimate of carbon flux, our daily
values of an average of 2.88 g C $m^{-2}$ $d^{-1}$ during the growing season, combined with the
relatively short dormant season relative to other temperate locations, suggest a high carbon
sequestration rate for this ecosystem type. In another southern hemisphere study, growing
season rates at an EC tower site in Argentina, are extrapolated by us to average 1.6 g C $m^{-2}$ $d^{-1}$
(Bautista et al., 2023) but in that saltmarsh, flooding reduced vegetation biomass and
productivity.

The data presented here is the exchange of carbon between the land surface and the
atmosphere, but saltmarshes, like other marine connected communities, exchange carbon also
through dissolved carbon pathways, which can be significant (Cai, 2011). Thus, the fluxes
presented here do not constitute the entire carbon budget of this ecosystem.

## 5 Conclusions

The response of the French Island saltmarsh to environmental drivers is indicative of the
complex interactions determining saltmarsh productivity. The unique long-term, high-
resolution record enabled us to derive temperature, VPD and light response functions, thus
formulating equations that describe how climate-change sensitive parameters such as
temperature, relative humidity, and cloud cover, affect $CO_2$ uptake, respiration and
evapotranspiration. The marsh operated as a $CO_2$ sink throughout the various canopy
phenological phases, but during the dormant period, $CO_2$ uptake was less than 25% that of
the growing season. Seasonality of greenhouse gas fluxes in Australian saltmarshes is an
understudied but important aspect of global carbon budgeting.

Competing interests

The contact author has declared that none of the authors has any competing interests.

Acknowledgments

The work was carried out with the permission of Parks Victoria (Permit 10008684). We thank
Phil and Yuko Bock for logistic support and accommodation on French Island. We thank
Leigh Burgess, Kiri Mason and Ian McHugh for technical support and the Australian OzFlux
community for ongoing collaboration. This work was funded by an Australian Research
Council Discovery Award to RR and ED (DP220102873) as well as a Monash University
Networks of Excellence award to RR.

Data Availability
Data used for this analysis is available at https://figshare.com/s/ba62aafd1a4049248a08 (note
that this is a temporary private link to an embargoed dataset which will be replaced with a
publicly available DOI upon publication).

Author contribution
RR conceptualised the study, acquired funding, prepared the manuscript, designed and
carried out the field campaign, and performed the analysis. ED acquired funding, developed
methodology and prepared the manuscript. AG developed methodology and prepared the
manuscript. TA, EJVH, HR and MP were involved in the field investigation and
administration of the project and provided edits on the manuscript.

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
