# Peer review of "Eddy Covariance Evaluation of Ecosystem Fluxes at a Temperate Saltmarsh in"

_EGUsphere, 2024_

## Author Response (AR1)

Thank you to the two reviewers for their comments. Kindly find detailed below our response to the peer review. The reviewer questions are indicated in bold and our answer reflect changes we have made to the manuscript and figures.

Please let me know if any clarification is required

Ruth Reef (on behalf of the co-authors).

**Review "Eddy Covariance Evaluation of Ecosystem Fluxes at a Temperate Saltmarsh in Victoria, Australia Shows Large CO2 Uptake" egusphere_2024-2182 by Reef et al.**

Reviewer 1

**Summary:**

The manuscript presents the first study using EC to measure CO2 exchange in an Australian temperate saltmarsh over nearly 2 years, from November 2019 to August 2021. CO2 fluxes over the saltmarsh were highly seasonal with greatest fluxes during the growing period and smallest fluxes during the dormant period. Investigated relationships with selected environmental drivers showed growing season daytime NEE was mostly dependent on radiation (short-wave incoming) followed by temperature (showing a temp optimum for CO2 uptake) and a threshold for VPD. Data coverage was compromised due to many factors and data was not gap-filled. Authors cautiously present an annual estimate for NEE (or NEP).

The study is certainly of interest and within the scope of Biogeosciences. Although the study is more descriptive it is novel being the first study of EC-measurements in a Southern Hemisphere/Australian temperate saltmarsh. Data coverage was compromised but this is often the nature with EC-measurements, hence gap-filling procedures are often crucial. The manuscript would need improvements on various aspects listed below and needs to address inconsistencies, and provide clarification, better descriptions and more rigorous approach around data processing and analysis.

**Q: Definition of season: Throughout the manuscript the definition and length of growing and dormant season is inconsistent.**

A: We are humbled to note that there was some confusion in the manuscript around the growing season terminology, but this confusion does not extend to the data analysis which followed the following condition:

Growing season:

>FI_week$GrowingSeason<-ifelse(FI_week$MonthofYear %in% c(6,7,8,9),0,1)

Thus the Dormant season was June through September, the rest was considered growing. The confusion stemmed from the initial use of multiple phases within the growing season ('greening up' etc...) , which we eventually collapsed into a single growing season period and were not careful enough in the manuscript to correct this. This has now been amended throughout the manuscript.

**Q: The authors acknowledge well the circumstances that led to data loss but it would be good to present a better table or figure on data gaps/availability in percentage. Currently, Fig 3) does not show much and is difficult to read with the current ratio of panels.**

A: Thank you for the suggestion, Table 1 has been added to the manuscript, detailing the number of daily and nightly half hourly datapoints per month as well as the mean and SD of the $CO_2$ flux for each period.

**Q: L237/238 - are the authors referring to half-hourly measurement points? The numbers themselves do not mean much, it would be better to state the percentage of data coverage during the selected measurement period, either as a whole or split by dormant and growing season (i.e. 1 y 9 m is roughly 638 days, i.e. data coverage is xx days or xx %)**

A: Table 1 now supports this text to better clarify the data available.

The authors need to describe in better detail what data processing steps have been done, i.e. for each correction are different options available and these should be stated and references cited with them. Current version L225-228 is insufficient. Furthermore, how as the u* threshold determined. This needs to be better described and clarified.

**Q: Why did the authors choose to not gap-fill data. While using non-gapfilled data is valid to explore relationships with environmental drivers, it makes strong outcomes of seasonal variability difficult if the data gaps are skewed towards certain periods of the day or year. Gap-filling bigger data gaps is problematic but gap-filling between time periods with most coverage should be possible and within the flux community sufficient approaches are available.**

A: The decision not to gap fill was to ensure that the main focus of the study – relationship between NEP and environmental drivers – could be assessed. Given that this is admittedly, a patchy dataset with large gaps, combined with the low amount of existing knowledge from this ecosystem, we opted not to gap fill. In cases of days with high coverage (i.e. days with data density of >85%) we fitted curves for

flux integration, which act as a gap-filling measure. These are the days we used to compare among seasons to avoid the skew mentioned by the reviewer.

**Terminology:**

**Q: NEP is the Net ecosystem productivity and is not equal to net ecosystem uptake, only when NEP is positive. The authors should clearly define and be consistent throughout the manuscript what negative or positive NEP is, either uptake or emission. Generally net ecosystem $CO_2$ uptake as NEP is positive, while shown as NEE it is negative. Most of the time positive NEP in the manuscript is uptake but then in the discussion NEP uptake off other studies are indicated with negative symbol and also the cited net emissions (L488 to L493).**

A: We defined NEP uptake as positive. However, in the literature some studies do not follow this general rule, and we maintained the sign used in those studies. We agree that a common terminology among studies would be highly beneficial.

**Q: L127 – instead of photosynthetic flux of $CO_2$ it would be better to say GPP of saltmarshes is the $CO_2$ uptake by all plants via photosynthesis. While GPP is explained, ecosystem respiration is not, hence it should be stated all potential $CO_2$ sources contributing to ecosystem respiration from the saltmarsh.**

A: We have amended these two sentences as suggested.

**Q: L268: not once is GEP mentioned further in the manuscript and thus, the definition here is not relevant.**

A: GEP has been removed

**Relationships with environmental drivers:**

Earlier studies have identified that NEE varies between saltmarshes that was linked to differences in species, salinity, hydrology and biochemical conditions. Authors should consider discussing highlighting such differences with regards to measured NEE in this study and reported ones in previous studies.

**Q: The authors mention that the saltmarsh experiences semi-diurnal tides. This is an important environmental driver for the ecosystem and consequently for $CO_2$ fluxes and thus it would be important to assess how a**

**change in water table would influence CO$_2$ exchange. I'm surprised that authors did not consider this in their study.**

A: The region experiences semi-diurnal tides and we did indeed consider this in our study. However, the hydrology of this marsh turned out to be very complicated due to non-tidal ponding behaviour and a shore front sand dune that limited the connectivity of the marsh with the sea. We completed a few rounds of water level logger deployments but frequently found that water level at the site did not correspond with tides and, due to micro topographical differences was not uniform across the site. Rain also resulted in ponding and often took 3-5 days to drain. Thus, while we agree that the water table matters, we could not assess this in this footprint due to limitations in assessing water level. We have recently moved our tower to a site that is lower in the intertidal and combined it with wells within the footprint to assess this question.

**Q: Authors mentioned heatwaves or high temperatures – did authors consider how daily or diurnal NEE patterns compare between different temperature conditions, i.e. hot days vs mild/warm days?**

A: We did, however there is quite a bit of complexity to this, for example how warm nights are affect diurnal NEE patterns too. We think that this extended analysis is outside the scope of this paper. The outcome of this analyses would add significant length to this paper, which focuses on seasonality, and we chose to work on this separately and across different climate zones.

**Q: Can authors clarify why they chose certain environmental drivers and correlations, i.e. Fig 5b – the rationale for the NEP – temperature relationship?**

A: The environmental relationships selected are based on common plant physiology forcings that display variability over the time scales studied here. The one omission is soil salinity, for which we did not have data. Functions to describe the relationships between environmental drivers and NEP were selected based on

**Q: I would like to see the relationship between ecosystem respiration and temperature that the authors mention but not show, as this is one of the most important relationships used in a method to estimate gross fluxes, GPP and ER, from NEE. This relationship is most certainly not linear, but exponential – commonly described with the Lloyd & Taylor type Arrhenius function (also see L266)**

A: The respiration/temperature relationship has been added as Fig. 6, the Lloyd&Taylor Arrhenius function did not fit the points better than the linear model suggested ($R^2$ = 0.09) but we have now used this relationship to correct for temperature effects on $R_{eco}$ to better align with the community.

**Q: Given the Gaussian relationships shown for temperature (or VPD), it would be good to explore and look into potentially confounding effect of environmental drivers as well.**

Temperature response in plants is Gaussian (parabolic) due to biological considerations rather than confounding effects of environmental drivers, C3 plants typically have an optimum range of 20-25C (Berry and Björkman 1980).

Berry, J., & Björkman, O. (1980). Photosynthetic response and adaptation to temperature in higher plants. *Annual Review of Plant Physiology*, 31, 491-543.

**Annual Budget:**

**Q: Given the substantial data gaps and the decision to not gap fill the data, presenting an annual NEP estimate, even cautiously, is not warranted. It is not where along the Nov 2019 to Aug 2021 timeline the data coverage is.**

A: The annual estimate has been removed, while we were not comfortable with providing one, we added it as we thought people citing this paper might attempt to create an annual budget from the data, and we wanted to provide a more cautious estimate.

**Q: It is not clear or understandable what is meant with the 24-hour carbon balance integration, for what period of time, what data? From Fig.6 it seems only the 90 + 18 days have been some sort of gap filled that should describe the daily range of NEE?**

A: The integration followed a mathematical area under the curve method, which is now detailed. We only included days where the data coverage is very high (>85% of the time points were retained following QC).

"To calculate the daily-integrated $CO_2$ and $H_2O$ fluxes, the daily sum of these fluxes was determined for days with at least 85% data coverage. This involved using the trapezoid rule to estimate the area under the curve for each of these 24-hour periods. The trapezoid rule approximates the total flux by dividing the day into smaller intervals, each lasting 1,800 seconds. For each interval, the area is calculated by averaging the flux values at the beginning and end of the interval, then multiplying by the interval duration. These areas are then summed to obtain the total daily flux. This method ensures that even with some missing data points, a reliable estimate of the daily flux can be obtained."

**Q: If only a certain days are getting gapfilled why not others? As mentioned above there are approaches available to gap fill data, either using monthly diurnal values or ANN, etc. However, it is not clear how the annual budget was calculated. It is not possible to derive an annual budget based on 108 days.**

**Thus, it is not possible to follow the steps that would allow any reproducibility.**

A: The annual estimate has been removed

**Q: The same applies to the data analysis in general. Authors state all has been done in R, but do not cite any significant or important package used, or R for itself.**

A: Thank you for noting this. Most of the analysis is self-scripted, but the packages used are now mentioned. The code is available on our Figshare site. This now reads:

"All post-processing and statistical analyses were performed in R 4.3.2 (R Core Team, 2024), including the packages *ggplto2*, *clifro*, *MASS*, *dismo*, *amerifluxr*, *rmarkdown*, *geosphere*, *ggmap* and *gbm*. "

**Q: I recommend to authors to look into better gapfilling procedures if some sort of annual budget is desired and also to improve the environmental driver analsyis. If the entire data analysis is only based on 90 or 18 days, it needs to be acknowledged that the significance of results is limited.**

A: An annual budget was removed from the output.

**Minor comments:**

**Q: The highlighted background colours are not explained (Fig2a). Does it show data gaps, availability or growing/dormant season?**

A: The caption now clarifies the background colour represents the growing and dormant season.

**Q: Fig 5 – better figure caption is necessary and should include what data has been used, i.e. daytime data, and only the growing season, non-gapfilled, but are these half-hourly or daily values?**

A: The caption has been corrected to detail "The relationship between growing season daytime half-hourly values of net ecosystem productivity (NEP, $\mu$mol $CO_2$ m$^{-2}$ s$^{-1}$)..."

**Q: In the text it states daytime NEE which has been set to >12 W .m-2 but the figure clearly shows data below that threshold. What is actually shown and the correlations based on?**

A: 12W m-2 is the *global* radiation value below which EddyPro considered the flux measurement to occur at night-time. The light response curve fitted to our day-time data is to the *net* shortwave radiation value. Net shortwave radiation is less than the total incoming solar radiation (but better reflects the radiation available to plants), which is why the x-axis extends below 12 W m$^{-2}$.

**Q: Authors should clarify what is uptake, i.e. positive NEP, but then the data also includes negative values, hence shown data it is not strictly uptake, but rather NEP – Net ecosystem productivity.**

A: We changed the word uptake to 'net ecosystem productivity' in the caption, to avoid confusion.

**Discussion:**

**Q: L402 to 411 – I question the relevance of non-flux processes with regards to seasonality. This is not really relevant for the carbon or CO$_2$ flux processes investigated here nor the defined objectives of the study.**

A: Given the novelty of the reporting of seasonality in temperate marshes in Australia, we find that this short discussion is relevant to this ecosystem.

**Q: In relation to NEE estimates from saltmarshes elsewhere, the discussion would improve to higlight the potential differences between these ecosystems that might explain differences in NEE (or NEP).**

A: We agree that this could be enhanced, and the discussion now includes potential differences, including differences in species composition, photosynthetic pathways and climate.

**Q: In the introduction it is highlighted that saltmarshes are defined by a strong seasonality but then in the discussion and conclusion it is somehow a novelty that Australian saltmarshes have a seasonality. This is confusing, can the authors explain in more detail?**

A: Australian saltmarshes are under studied. Whilst all publications from Australian marshes describe them as not having seasonality (e.g. Clark and Jacoby, 1994), or having seasonality that is based on rain events () none show any data relating to vegetation phenology. Despite this, studies on carbon cycling in these systems make the assumption that seasonality does not occur (Owers et al. 2018). In this study, we investigated the marsh over a period of a few years and are making an important point that like all other temperate marshes, Australian marshes (at least in the

south of the country) exhibit seasonality that is similar to those of other geographies.

**Q: ET – the authors acknowledge the distinct surface areas within the footprint that will affect any measurements on ET and thus estimates on WUE. These are also visible in Fig. 1c) and 1d) – Hence ET values, especially during the summer period might include a greater contributions of water bodies and thus would bias any WUE estimates currently presented. It might be possible the authors attempt to account for or sector out water bodies by i.e. excluding certain wind-sectors were water bodies are from the footprint .**

A: During the summer period, the marsh is relatively dry, and the saltpans ('water bodies') do not hold water. The marsh inundation regime is complex, and includes seasonal variation in inundation, with greater water accumulation on the surface during winter (and thus perhaps a smaller contribution of E to ET in summer). Thus, while the soil might be inundated in parts of the footprint, the complex hydrology results in a mosaic of emergent and submerged surfaces. We present the WUE at the ecosystem level, and do not partition evaporation from canopy evaporation.

Reviewer 2

**General comments:**

This manuscript is a substantial contribution to Biogeosciences because it contributes data from an ecosystem with limited historical observations. The methods are mostly valid, although there is some concern about the partitioning method, and missing references to recent, related work. Results are presented in a well-structured way, although there could be some rephrasing and improvement to figures to make reading easier.

**Specific comments:**

**Q: Page 2, Line 49: The abstract could better highlight the findings of this study, especially as it relates to environmental drivers.**

A: This has been improved

**Q: Page 2, Lines 38-39: There has been at least one other study of eddy covariance CO2 and CH4 fluxes from an Australian salt marsh (Safari et al., 2020).**

A: Safari et al. present eddy covariance data collected in a mangrove wetland in Tamago. While saltmarsh is present at the site, no footprint analysis is presented so it is unclear whether it was sampled.

**Q: Page 5, Line 140: While this was true in the past and historical data limitations in the Southern Hemisphere do exist, eddy covariance flux measurements have been and are being made in the Southern Hemisphere (Bautista et al., 2023). There is a growing effort to quantify wetland fluxes in the Southern Hemisphere, and new enthusiasm for collaborative programs like MexFlux that are drawing attention to flux science going on**

in Latin America (Vargas & Yépez, 2011). I suggest rephrasing this, perhaps to, "Previous EC studies in coastal saltmarshes have been largely focused on the Northern Hemisphere… but interest in data collected in the Southern Hemisphere is growing (sources)."

A: Thank you for the references, while Mexico is not in the southern hemisphere, the work of Vargas is, and is cited in our manuscript in four separate studies. Thank you for pointing to the Bautista manuscript, we have now included it (it was not published when we first submitted this manuscript).

**Q: Page 10, Line 263: What are the potential issues that could arise by using air-temperature corrected nighttime NEE as Reco? Could an artificial neural network or other machine learning algorithm be used to partition NEE in this dataset, despite the gaps? Suggest adding some information on the limitations of doing this to the discussion section.**

A: The nighttime NEE data is a common estimator for Reco, the temperature correction has been changed to follow the Lloyd & Taylor Arrhenius model rather than a linear regression. Errors due to low turbulence at night were avoided due to our u* filtering process.

**Q: Page 11, Lines 284-291: Can the font size of the x and y-axis labels and ticks in Figure 2A be increased for visibility? Also, the green and red shading doesn't quite align with the growing season and dormant period. Is that what the colors are supposed to represent? Please add this information into the description.**

A: Corrected

**Q: Page 18, Line 395: Something that is missing from this discussion that could be interesting to analyze or mention is the relationship between VPD, ET, and NEP at this site. If you were to plot half-hourly NEP corresponding to a VPD greater than 2 kPa, and color code the datapoints according to ET, would you find that ET is reduced during those times? This could lead to an interesting (but short) paragraph on ET reductions due to stomatal closure when VPD is high. The discussion could be combined with or added after the paragraph on ET (Line 444).**

A: We have added a colour scale based on ET to the figure, and find that low NEP at high VPD is related to a slight reduction in ET. We added a section to the discussion, as suggested here:

"The parabolic relationship between NEP and air temperature and NEP and VPD suggest that higher air temperatures and VPD (which are expected with climate change) could negatively impact $CO_2$ uptake by these coastal ecosystems. High VPD was related to lower NEP, and to a lesser extent, lower ET (Fig. 6d). However, VPD increases atmospheric demand for water, increasing the evaporation from the saturated marsh surfaces in the footprint, and this atmospheric demand could be forcing ET at high VPD rather than plant moderation via reduced transpiration, even if transpiration is reduced. Thus, despite maintained ET

during VPD periods we cannot conclude a non-closure of stomata. NEP also reduced below a VPD of 1.92 KPa, but at our field site low VPD correlated with low temperatures (r = 0.88), and low temperatures were shown to limit NEP."

**Q: Page 22, Line 503: The conclusions of this study are somewhat understated here. This study was unique in that it provided high-frequency measurements of an abundant greenhouse gas (CO2) using a precise technique (eddy covariance flux) in an ecosystem with limited historical measurements (coastal Australia). I suggest briefly restating the study methodology and highlighting the key findings and discussion points here: time series analysis was performed on CO2 flux measurements across various scales (daily, nightly, diel, half-hourly, hourly, seasonally) to assess the impacts of ET, SW Rad, VPD, and Tair and how those relationships change throughout the year. Unsurprisingly, CO2 uptake during growing season was higher. Dormant season CO2 uptake peaked later in the day. The relationships with NEP were Gaussian for air temp and VPD, linear for ET, and logarithmic for SW rad. This means that while plants appear to adjust to changes in ET and SW rad, higher air temps and VPD (which are expected with climate change) could stifle CO2 uptake by coastal ecosystems. This is a super important finding with climate modeling implications.**

A: Thank you! We have improved the discussion

**Technical corrections:**

**Q: Page 2, Line 53: Was is the primary intertidal vegetation referenced in the sentence "saltmarshes…are the primary intertidal vegetation"? Are saltmarshes the primary ecosystem or wetland type? Or are grasses and sedges the most common intertidal vegetation cover outside the tropics? Clarify or rephrase.**

A: The sentence was changed to read: Saltmarshes constitute 30% of these ecosystems globally and are the primary intertidal coastal wetland habitat outside the tropics

**Q: Page 8, Line 222: Remove comma after "The analysis"**

A: corrected

**Q: Page 9, Line 233: Add abbreviation at first appearance of the word. "…standard deviations (SD) from the mean…"**

A: corrected

**Q: Lines 333, 345, 348: Replace "standard deviation(s)" with "SD(s)"**

A: corrected

**Q: Page 12, Line 309: Suggest changing to "daytime and nighttime" for consistency with the rest of the paper**

A: corrected

**Q: Page 16, Line 362: Introduce the abbreviation at the first appearance of the word. Change caption to read: "Shortwave (SW) radiation…"**

A: corrected

**Q: Page 18: Would relabel y-axes to read "Growing season days" and "Dormant season days".**

A: corrected

**Q: Page 19, Line 399: Seasonality in Australian saltmarshes has been examined at least one other time (Safari et al., 2020). Suggest rewriting this sentence.**

A: Safari et al. (2020) describe a unique eddy covariance study in an Australian wetland, of combined mangrove and saltmarsh, and find an effect on rainfall periods on flux. However, they do not investigate the role of astronomical season.

**Q: Page 19, Line 407: Add hypen and add the word "for": "along the East Asian-Australasian Flyway, particularly for waders, thus…"**

A: corrected

**Q: Page 20, Line 431: Similar to earlier comments, there is existing data from Australian salt marshes.**

A: The Safari et al. (2020) paper describes a mangrove wetland rather than a saltmarsh.

**Q: Line 22, Line 497: Change to "estimated by Krauss et al. (2022) to average…"**

A: corrected

**Q: Page 22, Line 514: Suggest rephrasing based on previous comments to something like "Seasonality of greenhouse gas fluxes in Australian salt marshes is an understudied but important aspect of global carbon budgeting…"**

A: corrected

**References:**

Safari, D., Edwards, G. C., & Gyabaah, F. (2020). Diurnal and Seasonal Variation of CO2 and CH4 Fluxes in Tomago Wetland. International Journal of Sciences, 9(01), 41-51.

Vargas, R., & Yépez, E. A. (2011). Toward a Mexican eddy covariance network for Carbon Cycle Science.

Bautista, N. E., Gassmann, M. I., & Pérez, C. F. (2023). Gross Primary Production, Ecosystem Respiration, and Net Ecosystem Production in a Southeastern South American Salt Marsh. Estuaries and Coasts, 46(7), 1923-1937.

---

## Author Response (AR2)

Dear editor, kindly find below my responses to the reviewer comments received.
Thank you for the handling of our paper, it has been an excellent process
Ruth, on behalf of the co-authors.

Detailed comments:
L183: check the subsection numbering, it shows 2.82 instead of 2.1, check throughout the entire text, i.e. L231 etc.
FIXED
L196: not sure what the track change is meant to be here – worth checking
I'M NOT SURE WHAT THE FORMATTING ISSUE IS, BUT THE TEXT IS AS WHAT I EXPECT
L223: check track changes – it looks like some formatting error
AS ABOVE
L254: can authors maybe add a citation or link to the table for the Express Mode settings of Eddy Pro
I'VE PROVIDED THE LINK IN THE TEXT
L273: It is not clear if the authors applied a common method to assess the friction velocity threshold or if they applied a threshold value "typically used in EC studies" (citing a study from a 447m tall tower) – which is itself not a correct statement. A friction velocity threshold needs to be determined for each site individually and can range from 0.1 to 0.5 m.s-1 or higher (see Chapter 5 in Aubinet, M., Vesala, T., and Papale, D. (Eds.): Eddy Covariance, Springer Netherlands, Dordrecht, https://doi.org/10.1007/978-94-007-2351-1, 2012.).
The widely applied method now to determine u* thresholds is the Change Point Detection method after Barr (2013):
Barr, A. G., Richardson, A. D., Hollinger, D. Y., Papale, D., Arain, M. A., Black, T. A., Bohrer, G., Dragoni, D., Fischer, M. L., Gu, L., Law, B. E., Margolis, H. A., McCaughey, J. H., Munger, J. W., Oechel, W., and Schaeffer, K.: Use of change-point detection for friction-velocity threshold evaluation in eddy-covariance studies, Agricultural And Forest Meteorology, 171, 31–45, https://doi.org/10.1016/j.agrformet.2012.11.023, 2013.
I'VE CHANGED THE SENTENCE TO HIGHLIGHT WE USED THIS METHOD AND ADDED THE REFERENCE. THE REFERENCE I USED EARLIER WAS TO THE METHOD NOT THE NUMBER, BUT IT WAS NOT CLEAR TO THE READER.

L276 & Table 1: It's good to see the table of data coverage which makes the study and more transparent. However, it would be good to show where the 90 days for growing season and 18 days for dormant season data is coming from. Given the widespread intermittent coverage, it's not clear from which time period (i.e. month and year) these explicit selected days are coming from. This related to L467: "reflecting several phenological stages" – given the variability, it might be good to show a boxplot or depending on the spread of the 90 days across several phenological stages – it might be interesting to see differences in daily NEE corresponding to phenological stages.

I'VE ADDED THE SEASONS AS A SPEARATE COLUMN (AND REMOVED COLOUR). UNFORTUNATELY AT THIS POINT I DON'T THINK THERE'S ENOUGH DATA TO BREAK IT DOWN FURTHER TEMPORALLY, BUT THIS WOULD BE AN AREA OF INTEREST FOR ME.

L312: 1800 seconds which is 30 min, i.e. as data interval.

NOW READS: 'The trapezoid rule approximates the total flux by dividing the day into smaller intervals, each lasting 1,800 seconds (30 minutes). For each data interval…l'

L320: GPP instead of GEP, the paragraph is a bit confusing and needs clarification, see below.

CHANGED GEP TO GPP

L387: better to say night time only ecosystem respiration.

Paragraphs starting L320 and L387: There is the seeming assumption that night-time RE is the same as day-time RE. While mentioned earlier that partitioning was not possible, this is a partitioning (the night-time approach using night-time RE and temperature relationship), i.e. calculate RE based on the temperature relationship, so to get estimates for the daytime (based on daytime temperature) which then get subtracted from NEE to get GPP. RE is usually higher during day-time than night-time due to higher temperatures, which is also reflected in the seasonal changes, i.e. RE is higher in summer than winter due to temperature. Thus, taking nighttime RE to

get a daytime GPP estimate is problematic. It would be better to exclude the part relating to GPP (or GEP), as it is not further discussed.

I HAVE REMOVED THIS SECTION, IT NOW READS "For the $CO_2$ budget, Net Ecosystem Production (NEP), was defined as NEP=-NEE. Nighttime NEE is referred to as $R_e$ and was corrected for temperature effects on respiration using a an exponential Arrhenius-type relationship (Lloyd and Taylor, 1994). "

Admitting the RE-temp relationship is not good; Why has the data been selected between 22:00 and 2:00 am? Did authors used the u* filtered nighttime NEE and/or tried to use only the first 3 hours after sunset (i.e. van Gorsel method, van Gorsel et al., 2007) to see if the relationship improves?
van Gorsel, E., Leuning, R., Cleugh, H. A., Keith, H., and Suni, T.: Nocturnal carbon efflux: reconciliation of eddy covariance and chamber measurements using an alternative to the u(*)-threshold filtering technique, Tellus B: Chemical and Physical Meteorology, 59, 397–403, https://doi.org/10.1111/j.1600-0889.2007.00252.x, 2007.

WE SELECTED THE DATA TO BE UNIFORM THROUGHOUT THE YEAR AND SUMMER DAYS ARE VERY LONG. DUE TO THE HIGH LATITUDE LIGHT LEVELS AFTER SUNSET (TWIGHLIGHT) ARE HIGHLY VARIABLE ACROSS SEASONS SO OUR APPROACH WAS SELECTED TO ENSURE DARKNESS.

L329: please correct instead of "a linear slope of the relationship": "… an exponential Arrhenius-type relationship (Lloyrd & Taylor, 1994) …"

CORRECTED AND REFRENCE ADDED

L335-336: better to remove the confounding phrase "which has a relatively short growing season during the summer" if the growing season is Oct-May

REMOVED

L355: Thanks for the season clarification. It would be better to just colour the growing season as somehow the colouring of the seasons in the figure does not always seem to align with Oct-May and June-Sep.

IN THE ORIGINAL IMAGE I'VE ONLY COLOURED THE PERIODS WE ARE USING DATA FROM, NOW I HAVE UPLOADED A NEW FIGURE COLOURING ALL PERIODS BY SEASONS.

Fig.3: It would be good to improve this figure, i.e. splitting each panel into years (2020 & 2021), to avoid this big gap in between data point clouds – that way the data will be better visible.

FIGURE 3 WAS CHANGED AS PER RECOMMENDATIONS

L426 & Fig. 6a: This is in a way the Lasslop approach for NEE partitioning using day-time data with the y-intercept is the respiration value for daytime – something to consider for the authors.

THANK YOU

L505: be mindful of consistency of acronyms, i.e. Tair or TA or airT, same with SW radiation or Rad,…

FIXED

L512: instead of carbon budget use daily carbon fluxes

FIXED

The authors explained well in the reviewer response why the changes in the water table were not considered, which would be important to briefly include in the material and methods.

FIXED, now reads "Western Port has semi-diurnal tides with a range of nearly 3 m, resulting in wide intertidal flats occupied by mangroves of the species *Avicennia marina* and saltmarshes. The saltmarsh in this study experiences complex hydrological conditions, and we found that inundation does not directly link to tides. "